# Geographic variability of dust and temperature in climate scaling regimes over the Last Glacial Cycle.

Nicolás Acuña Reyes[1], Elwin van 't Wout[1], Shaun Lovejoy[2], and Fabrice Lambert[3]

[1]Institute for Mathematical and Computational Engineering, School of Engineering and Faculty of Mathematics, Pontificia Universidad Católica de Chile
[2]Physics Department, McGill University, Canada.
[3]Geography Institute, Pontificia Universidad Católica de Chile

**Correspondence:** Fabrice Lambert (lambert@uc.cl)

**Abstract.** Temperature and mineral dust records serve as valuable paleoclimatic indicators for studying atmospheric variability across different temporal scales. In this study, we employed Haar fluctuations to analyse global spatio-temporal atmospheric variability over the last glacial cycle, capturing both high- and low-frequency information within the records, regardless of uniform or non-uniform sampling. Furthermore, we utilised Haar fluctuations to compute fluctuation correlations, thereby enhancing our understanding of paleoclimate dynamics.

Our findings reveal a latitudinal dependency in the transition from macroweather to climate regimes ($\tau_c$), with polar regions experiencing shorter transitions compared to the tropics and mid-latitudes. These transitions occur at approximately 1/100th of glacial cycle length scales, suggesting a dominant forcing mechanism beyond Milankovitch cycles. Additionally, our analysis shows that polar regions have larger fluctuation amplitudes than lower latitudes as a consequence of the polar amplification effect. Furthermore, fluctuation correlations demonstrate faster synchronisation between the poles themselves compared to lower latitude sites, achieving high correlation values within 10 kyr.

Therefore, our findings suggest a consistent climate signal propagating from the poles to the equator, representing the first empirical evidence supporting the hypothesis that the poles play a pivotal role as climate change drivers influencing the variability of climatic transitions worldwide.

## 1  Introduction

The study of paleoclimate, which involves reconstructing past climate variations using various paleoclimatic archives, provides crucial insights into understanding Earth's climatic history (Ruddiman, 2008). Paleoclimatic archives, such as ice cores and sedimentary records, offer unique opportunities to investigate climate dynamics on various timescales, ranging from annual to millennial and beyond (North Greenland Ice Core Project members et al., 2004; Lisiecki and Raymo, 2005). These archives preserve information about past changes in relevant climate variables such as air and sea temperatures, atmospheric aerosol content, and precipitation regimes. This enables us to discern the patterns and drivers of climate variability over extended periods (von der Heydt et al., 2021).

Understanding the temporal complexity of paleoclimatic records has been a long-standing challenge in climate science. The inherent non-linear and chaotic nature of Earth's climate system often leads to multifaceted interactions and feedback mechanisms acting over a wide range of scales. Conventional statistical methods, although essential, may not fully capture the intricate dynamics within the paleoclimatic time series, leaving significant patterns undiscovered. Furthermore, because of the many different physical processes interacting within the climate system, its variability spectrum is far from being composed of pure spikes belonging to well-defined forcing processes, such as daily variations or Milankovich cycles. Instead, the bulk of the spectrum is continuous and scales over a wide range. Indeed, from fractions of a second to hundreds of millions of years, almost all of the variability resides in wide scaling regimes that for a long time had been reduced to an uninteresting "background" (Lovejoy, 2015). This extends by many orders of magnitude, the range of scales considered by Wunsch (2003) who showed that over the range of millennia to Milankovitch scales, that quasi-periodic signals represent only a small fraction of total variability.

The quantitative importance of the continuum part of the variability is slowly being recognized. A recent ad hoc "solution" proposed by (von der Heydt et al., 2021) is to redefine the background as a "$1/f$" noise instead of white noise. This redefinition allowed them to produce a "conceptual landscape" displaying interesting narrow-scale range processes above the new background level. Because the range of scales shown in the conceptual landscape was $10^{13}$, the redefinition effectively boosts the background of the high frequencies by the same factor. Although this (partially) offsets the factor of $10^{15}$ error in Mitchell's 1976 (Mitchell, 1976) "educated guess", as shown in Lovejoy (2023), the resulting boosted background is not yet representative of the empirical data. Indeed, the ad hoc assumption of a $1/f$ background constitutes an extreme scaling hypothesis that only approximates empirical measurements. As argued in (Lovejoy, 2015), one way to improve our analysis of the background spectrum is to investigate real-space fluctuations (instead of Fourier-space spectra), such as the Haar fluctuation.

Classically, atmospheric dynamics is divided into two basic regimes: weather and climate. Generally, the definition of the climate regime is the thirty year average proclaimed by the International Meteorological Organization in 1934, despite abundant data indicating long-term trends, particularly in the context of modern climate change. In a non-steady state, dynamical regimes should be objectively defined, and scaling is one way to achieve this. If we denote by $\Delta T(\Delta t)$ a fluctuation in the temperature over a time interval $\Delta t$, then in a scaling regime $\Delta T(\Delta t) \approx \Delta t^\zeta$. In a general (multifractal) case, $\zeta$ is a random exponent. To simplify the discussion, we average this relationship to obtain the average fluctuation $\langle \Delta T(\Delta t) \rangle \approx \Delta t^H$ where $H$ is the (non random) "fluctuation exponent" ( "$\langle \rangle$" denotes averages). If such a relation holds, then, to within the constant factor $\lambda^H$, fluctuations over a long time interval $\lambda \Delta t$, ($\lambda > 1$) are the same as those over a shorter time scale $\Delta t$: $\langle \Delta T(\lambda \Delta t) \rangle \approx \lambda^H \langle \Delta T(\Delta t) \rangle$ (Lovejoy, 2023; Franzke et al., 2020). Through systematic scaling analyses of atmospheric fluctuations over the range of milliseconds to millions of years, Lovejoy and Schertzer (2013) and Lovejoy (2015) suggested that atmospheric dynamics has five distinct regimes rather than two. In order of increasing time scale/duration, these are:

a) The weather regime:

This extends from turbulent dissipation scales (below 1s) to the lifetime of planetary structures (typically about 10 days in the atmosphere but about six months in the ocean). In this regime, (after removing the diurnal cycle), average fluctuations

$\langle \Delta T(\Delta t) \rangle$ have $H > 0$ and hence increase in amplitude with time scale revealing "unstable" temperature time series that tends to "wander" up and down.

### b) The macroweather regime:

Beyond the weather regime lies the macroweather regime, in which fluctuations tend to decrease with scale ($H < 0$). In this regime, successive fluctuations tend to cancel, and the temperature anomalies tend to converge to well-defined values. This is the "long term weather" regime that resembles the classical 30-year average "climate": by increasing the timescale the average converges more and more.

### c) The climate regime:

The macroweather convergence of anomalies does not extend beyond a critical time scale $\tau_c$, beyond which the apparent point of convergence starts to wander (again $H > 0$). In the Anthropocene, the situation is fairly straightforward: $\tau_c$ is the scale at which the temperature response to anthropogenic forcing exceeds the internal variability; $\tau_c$ is currently about 15 years. The variability of temperature fluctuations in the Anthropocene clearly shows the importance of $\tau_c$, a transition between macroweather convergence to a well-defined "climate state" and a new lower frequency regime of "climate change". The

example of the Anthropocene also provides insight into the causes of the transition: the emergence of new sources of variability, in this case primarily greenhouse gases. Whereas when using anthropogenic forcing, General Circulation Models (GCMs) can reasonably reproduce the current $\tau_c$, in control runs (fixed forcing), they show macroweather convergence to a well-defined climate state out to their longest simulated time scales (Lovejoy and Schertzer, 2013; Lovejoy, 2023). In the pre-industrial epoch, neither the transition time $\tau_c$ nor its origin is known. For example, high-resolution dust and temperature data from an

Antarctic ice cores suggest that it is of the order of several centuries to several millennia, varying from one interglacial to another (Lovejoy and Lambert, 2019). However, these time scales are too short to be consequences of Milankovitch forcing. Is there a new source of external forcing or is there a new non-linear internal forcing mechanism? There is also a reason to expect geographical variation in $\tau_c$, which is the object of this study.

### d) The macroclimate regime:

The wandering ($H > 0$) climate regime extends to Milankovitch scales ($\approx 100$ kyr) beyond which again, $H < 0$ and fluctuations decrease with scale and anomalies converge. Little is known about this regime although the negative H regime is thought to be a feature of paleoclimatic data between timescales of about 100 kyr and 1 Myr throughout the Phanerozoic.

### e) The megaclimate regime:

This H>0 regime extends from $\approx 1$ Myr to at least several hundred million years and seems to be fundamentally bio-

geological. Wandering paleotemperatures with $H > 0$ apparently coexist with converging ($H < 0$) extinction and origination rates over the entire range. Interestingly, fundamentally important biodiversity seems to be dominated up to about 40 Myr by the destabilizing climate, but beyond that by stability macrobiology (Spiridonov and Lovejoy, 2022).

Over the past decade, numerous studies have focused on scaling paleoclimatic data (Fredriksen and Rypdal, 2016; Laepple and Huybers, 2013; Lovejoy and Varotsos, 2016; Nilsen et al., 2016). Some studies have evaluated how well climate models

reproduce the observed long-term climate variability (e.g., Zhu et al., 2019). While climate models and reconstructions show reasonable consistency on seasonal to inter-annual time scales, significant discrepancies emerge at longer time and regional

scales, with models underestimating the variability (Laepple and Huybers, 2014; Hébert et al., 2022). The best-known variability at millennial time scales is related to Dansgaard-Oeschger (DO) events, which are abrupt temperature changes originating in the North Atlantic region. They influence the climate over the entire northern hemisphere (Svensson et al., 2008), and are

phase-lagged compared to their Southern Hemisphere counterparts, the Antarctic Isotopic Maximum (AIM) events (EPICA Community Members et al., 2006). They are mechanistically linked to variations in the Atlantic Meridional Ocean Circulation (Stocker and Johnsen, 2003). At multi-millennial time scales, the best-known variability is caused by external orbital forcing, which redistributes incoming solar radiation around the globe. The main periodicities are linked to the precession (∼19-21 kyr), obliquity (∼41 kyr), and eccentricity (∼100 and 400 kyr) of the Earth's orbit, also called the Milankovitch cycles. The

weakness of models to capture low frequency variability means that although weather and to a certain degree macroweather variability can be analyzed in simulation data, the longer climatic timescales and the transition between macroweather and climate can only be reliably investigated in empirical records. However, because of the limited availability of paleoclimatic records with high resolution and long duration, such studies typically focus on timescales shorter than glacial cycles (Rehfeld et al., 2018) and longer spatiotemporal patterns are not yet well understood. As a result, little is known about the geographic

differences in $\tau_c$ (the transition time scale between macroweather and climate) .

In this study, we compiled a list of paleoclimatic temperature and dust records that were measured at high resolution and span most of the last glacial-interglacial cycle. Temperature data are typically based on isotope measurements (Jouzel et al., 2003), whereas dust data are typically based on direct particle measurements (Delmonte et al., 2002) or elemental data (e.g. calcium (Fuhrer et al., 1993)). Dust particles are usually entrained in continental arid regions and then transported by prevailing

winds towards remote regions of the globe where they are deposited by gravitational settling or precipitation over the surfaces of oceans, lakes, ice sheets etc. (Maher et al., 2010). Unlike isotopes, dust particles measured in paleoclimatic archives are therefore not representative of a single atmospheric variable, but indicate a more holistic state of the atmosphere (hydrological cycle, atmospheric dynamics). Paleoclimatic proxy records usually have non-equidistant sampling rates and sometimes lose resolution as we move to older time sections, limiting high-frequency analyses. Preprocessing techniques such as linear

interpolation can convert non-uniform data resolution to a uniform time series, but this always implies some high-frequency information loss, which is essential for finding scaling transitions. In contrast, Haar fluctuations are convenient for quantifying variability as a function of time scale. This enables us to accurately estimate fluctuations and other scaling exponents (Lovejoy and Schertzer, 2012) even for data with highly non-uniform chronologies (Lovejoy, 2015). In this context, Haar fluctuations have been shown to be more robust and less biased than conventional spectral analysis (Hébert et al., 2021).

Here, we analyze the global spatio-temporal atmospheric variability over the last glacial cycle, using a variety of paleoclimatic temperature and dust records. We calculated Haar scaling for each site and estimated the macroweather-to-climate transition scale $\tau_c$ for various latitudinal ranges. Because climate involves many intimately coupled processes, understanding variability at any timescale also requires spatiotemporal analysis (Huybers and Curry, 2006). Hence, we also used our non-uniform Haar fluctuation to quantify the fluctuation correlations between different datasets as a function of the timescale.

## 2 Datasets description

For our analysis, we used paleoclimatic records of both temperature and dust. Records obtained from marine sediment cores, ice cores, loess deposits, and lake sediments were selected. Most sites have records of either temperature or dust, with only a few sites having both, which may have been measured at different resolutions. Because we are specifically interested in covering the macro-weather and climate regimes, our analysis mostly includes paleoclimatic archives that cover at least the past 50,000 years at centennial resolution or better. Figure 1 and Table 1 show the locations and characteristics of each record included in this study.

Most sites are located in polar regions, such as Greenland and Antarctica, where measuring proxies at a high temporal resolution is easier to achieve than in the tropics and mid-latitudes. Unfortunately, no datasets that satisfied our selection criteria were available for regions such as Central and South America, Central Asia, Australia, the South Atlantic and the South Indian Ocean. Some datasets include multiple glacial cycles, such as the EPICA Dome C records or the Xifeng loess archive, whereas others cover only a single glacial-interglacial cycle or less. To avoid mixing information from multiple glacial cycles, we restricted the analysis to the most recent 130 kyr, roughly the last glacial-interglacial cycle.

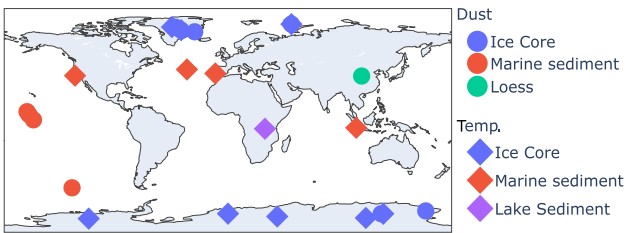

**Figure 1.** The global distribution of the datasets included in this study. The temperature and dust flux records come from ice cores, loess, lake and marine sediments.

Paleoclimate archives rarely have uniform resolution because each data point corresponds to a thin layer of core material, and each layer typically relates to a different time span depending on depth. Figure 2 shows the cumulative age covered by the sampling for each dataset. Note that a uniform dataset has a linear relationship, but most curves are above the diagonal, indicating a higher resolution in the early samples and a lower resolution in the older samples. We observe that the sampling rates depend on a specific dataset, and most are significantly non-uniform at all scales, that is the nonuniformity itself is likely to be scaling.

The datasets contain records associated with various dating scales during the last glacial period. We aimed to use the AICC2012 (Veres et al., 2013) or GICC05 (Svensson et al., 2008) dating scale on as many records as possible. Table 1 lists the specific dating scales used in this study. In Figure 3, we show all the time series over the most recent 130 kyr.

**Table 1.** Summary of the datasets included in this study.

| Name | Data id | Latitude (°) | Longitude (°) | Proxy type | Span (ky) | Resolution (y) | Samples | Timescale | Unit | Reference |
|---|---|---|---|---|---|---|---|---|---|---|
| | | | | Temperature measurements | | | | | | |
| WAIS | WAIS | -79 | -112 | Ice core | 68 | 11 | 6,375 | WD2014 | $\delta$O18 (per mil) | (Buizert et al., 2015) |
| Vostok | Vostok | -78.46 | 106.87 | Ice core | 423 | 128 | 1,853 | AICC12 | $\delta$D (%SMOW) | (Petit et al., 1999) |
| Dome Fuji | DF | -77.3 | 39.7 | Ice core | 340 | 286 | 1,189 | DFO2006 | $\delta$O18 (per mil) | (Sato et al., 2013) |
| EPICA Dronning Maud Land | EDML | -75.00 | 0.07 | Ice core | 150 | 65 | 2,259 | AICC12 | $\delta$O18 (%SMOW) | (Ruth et al., 2007) |
| Epica Dome C | EDC | -75.1 | 123.35 | Ice core | 801 | 138 | 5,787 | AICC12 | $\delta$D (per mil) | (Jouzel et al., 2007) |
| Lake Tanganyika - Eastern Africa | LT | -6.65 | 29.80 | Lake sediment | 59 | 280 | 209 | Local | °C | (Tierney et al., 2008) |
| Tropical Indian Ocean | Indian Ocean | -5.94 | 103.25 | Marine sediment | 131 | 629 | 210 | Local | °C | (Mohtadi et al., 2010) |
| Central California | California | 36 | -123 | Marine sediment | 130 | 544 | 214 | Local | °C | (Lyle et al., 2010) |
| Portugal | Portugal | 37.8 | -10.13 | Marine sediment | 162 | 357 | 330 | GICC05 | °C | (Tzedakis et al., 2020) |
| North Atlantic Ocean | Atlc.Ocean | 41 | -33 | Marine sediment | 69 | 363 | 178 | Local | °C | (Naafs et al., 2013) |
| GRIP | GRIP | 72.59 | -37.64 | Ice core | 104 | 20 | 5,200 | GICC05 | $\delta$O18 (‰) | (Rasmussen et al., 2014) |
| NGRIP | NGRIP | 75.1 | -42.3 | Ice core | 122 | 20 | 6,114 | GICC05 | $\delta$O18 (‰) | (Rasmussen et al., 2014) |
| | | | | | 122 | 20 | 5,651 | GICC05 | °C | (Kindler et al., 2014) |
| NEEM | NEEM | 77.45 | 51.06 | Ice core | 111 | 5 | 19,946 | GICC05 | $\delta$O18 (%SMOW) | (Gkinis et al., 2021) |
| Antarctic stack | Antarctic stack | - | - | Ice core | 140 | 10 | 13,980 | GICC05 | °C | (Davtian and Bard, 2023) |
| | | | | Dust measurements | | | | | | |
| EPICA Dome C | EDC | -75.10 | 123.35 | Ice core | 130 | 1 | 116,637 | AICC12 | Dust flux (mg/m$^2$/y) | (Lambert et al., 2012) |
| Talos Dome | TD | -72.82 | 159.18 | Ice core | 150 | 2 | 64,998 | AICC12 | Calcium ($\mu$g/kg) | (Schüpbach et al., 2013) |
| Central South Pacific | PS75 | -54.22 | -125.43 | Marine sediment | 474 | 199 | 2,384 | AICC12 | Lithogenic (wt-%) | (Lamy et al., 2014) |
| Central Pacific Ocean 1 | 17PC | 0.48 | -156.45 | Marine sediment | 149 | 756 | 192 | Local | Dust flux (g/m$^2$/y) | (A.Jacobel et al., 2016) |
| Central Pacific Ocean 2 | 31BB | 4.68 | -160.05 | Marine sediment | 141 | 531 | 254 | Local | Dust flux (g/m$^2$/y) | (A.Jacobel et al., 2016) |
| Central Pacific Ocean 3 | 37BB | 7.04 | -161.63 | Marine sediment | 144 | 1,319 | 105 | Local | Dust flux (g/m$^2$/y) | (A.Jacobel et al., 2016) |
| Xifeng - China | Xifeng | 35.70 | 107.60 | Loess | 800 | 1,108 | 722 | Local | Dust flux (g/m$^2$/ky) | (Guo et al., 2009) |
| RECAP | RECAP | 71.30 | -26.72 | Ice core | 121 | 66 | 2,317 | GICC05 | Dust conc ($\mu$g/kg) | (Simonsen et al., 2019) |
| GRIP | GRIP | 72.59 | -37.64 | Ice core | 104 | 20 | 4,806 | GICC05 | Calcium (ppb) | (Rasmussen et al., 2014) |
| NGRIP | NGRIP | 75.00 | -42.30 | Ice core | 108 | 20 | 4,869 | GICC05 | Calcium (ppb) | (Rasmussen et al., 2014) |
| NEEM | NEEM | 77.45 | 51.06 | Ice core | 108 | 10 | 9,736 | GICC05 | Calcium ($\mu$g/kg) | (Erhardt et al., 2022) |

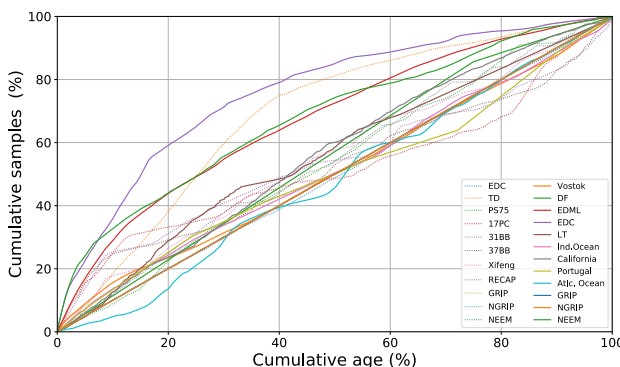

**Figure 2.** Cumulative age versus cumulative sampling, depicting the proportion of age covered by the proportion of sampling in the dataset, corresponding to the most recent 130 kyr of the datasets. The continuous lines depict temperature datasets while the dotted lines depict dust datasets.

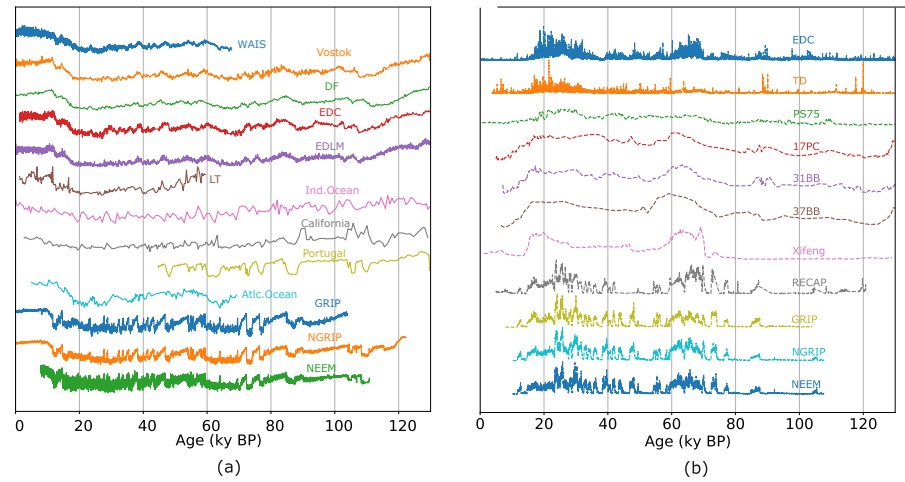

**Figure 3.** Temperature (a) and Dust (b) time series comparison for the last 130 kyr. See Table 1 for the ID of the records.

## 3 Methodology

This study aimed to analyze climate scaling regimes from paleoclimate data, which tend to have non-equidistant samples and many spikes. Non-uniform spectral estimators, such as Lomb-Scargle, have serious problems with uncontrolled spectral leakage, leading to low accuracy when applied to non-uniformly sampled scaling data. We applied a non-uniform FFT algorithm to the data but the scaling regimes were unclear from the analysis. Haar fluctuations are more robust on spiky data and better suited for identifying different scaling regimes. Furthermore, no preprocessing of the data, such as normalization and outlier filtering, is necessary for Haar fluctuation analysis (Hébert et al., 2021).


## 3.1 Haar fluctuations

Haar fluctuation analysis is a powerful technique for characterizing the timescale-dependent variability in a time series $F(t)$. The Haar fluctuation has a simple definition and the results have a clear interpretation. Furthermore, it can be extended to irregular sampling rates, thus avoiding data preprocessing such as interpolation between time samples. Hence, the algorithm uses all the information available from the data, including shorter timescales represented by high-resolution sections.

The Haar wavelet is a piecewise constant function that is used as a shape function in fluctuation analysis. The Haar fluctuation

$\Delta F$ over a uniform time interval $\Delta t$ is defined as the absolute difference in the mean over the first and second halves of an interval:

$$\Delta F(\Delta t) = \frac{1}{\frac{\Delta t}{2}} \left( \int_{t-\Delta t}^{t-\Delta t/2} F(t')dt' - \int_{t-\Delta t/2}^{t} F(t')dt' \right). \tag{1}$$

When the Haar fluctuation over a range of timescales $\Delta t$ obeys a power-law relation in the form of $\Delta F(\Delta t) \propto \Delta t^H$, the mean absolute fluctuation also varies as

$$\langle |\Delta F(\Delta t)| \rangle \propto \Delta t^H, \tag{2}$$

where the brackets "$\langle \rangle$" indicate the sample mean over all available disjointed intervals of size $\Delta t$ (fluctuation period) in the time series. Here, the exponent $H$ characterises the scaling regime, with $H > 0$ indicating that the average fluctuations increase with scale, while $H < 0$ implies the opposite. This fluctuation exponent estimation is appropriate within the range $H \in [-1, 1]$, which is valid for almost all geophysical data analysed to date (Lovejoy and Schertzer, 2012). A more general $q$-th moment of

the fluctuations can be estimated by defining the generalised structure functions $S_q(\Delta t)$ as

$$S_q(\Delta t) = \langle |\Delta F(\Delta t)|^q \rangle \propto \Delta t^{\xi(q)}, \tag{3}$$

where the exponent

$$\xi(q) = qH - K(q) \tag{4}$$

includes a linear term and a non-linear function $K(q)$ that characterises the multifractal scaling properties of the series $F(t)$.

This includes both the intermittency and extremes.

The definition of Haar fluctuation in Equation (1) assumes a uniform timestep. The extension of Haar analysis to non-uniform time series is detailed in Appendix A, following a similar approach as demonstrated in Lovejoy (2015, App. B).

## 3.2 Fluctuations Correlation

Thus far, climate variability has been described by scaling relationships in the form of $\Delta F(\Delta t) \propto \Delta t^H$ and calculated using

Haar fluctuations for a single time series only. If the governing processes have consistent scaling patterns, we should expect the interrelations between the processes to lead to correlations. The correlations between two time series at different scaling

regimes can be calculated using fluctuation correlation analysis. This technique estimates the normalised correlation coefficient ($\rho(\Delta t)$) between individual fluctuations at fixed time scales, that is, there is one correlation value at each $\Delta t$.

Let us consider two time series denoted as $A(t)$ and $B(t)$. The Haar fluctuations have the property that they are independent of any additive constant, therefore, $\langle \Delta A \rangle = \langle \Delta B \rangle = 0$. Furthermore, $\Delta A(\Delta t)$ and $\Delta B(\Delta t)$ denote the Haar fluctuations in $A$ and $B$, respectively. Then, at a time lag $\Delta t$ we can compute the fluctuation correlation, as follows:

$$\rho_{\Delta A, \Delta B}(\Delta t) = \frac{\langle \Delta A \, \Delta B \rangle}{\langle \Delta A^2 \rangle^{1/2} \langle \Delta B^2 \rangle^{1/2}}, \tag{5}$$

with the correlation coefficient $\rho_{\Delta A, \Delta B}(\Delta t) \in [-1, 1]$.

Because the time series likely have different sampling rates and durations, their fluctuations will cover time-scales that may not match each other, leading to an inaccurate estimation of $\rho$. To avoid this problem, we use linear interpolation to shift the sampling of the higher-resolution series, say $A(t)$, and fit it with the sampling of the lower-resolution series, say $B(t)$, before computing its fluctuations. Shortening the time-series duration to the maximum common $\Delta t$ is also required. Finally, we fitted a cubic spline to simplify the visualisation because we ended up with a cloud of fluctuations.

## 4  Results

This section presents the results of the Haar fluctuation analysis conducted on the selected datasets. Section 4.1 illustrates the scaling regimes observed at individual sites and provides a comparative analysis of the Haar fluctuation amplitudes. Fluctuation correlations are presented in Section 4.2.

### 4.1  Haar fluctuation analysis

The Haar analysis yields a fluctuation Root Mean Square (RMS) at each time lag so that the scaling exponent $H$ can be fitted for different regimes, as given by the Appendix Equation (A5). To analyze the spatial differences in the scaling regimes, we clustered the measurement sites into five different groups according to latitude.

- Arctic: From 90° to 60°.

- Northern mid-latitudes: from 60° to 20°.

- The tropics: From 20° to -20°.

- Southern mid-latitudes: From -20 to -60°.

- Antarctica: From -60° to -90°.

As explained in Section 3.1, when the average fluctuations increase with scale ($H > 0$), they represent unstable behaviour, whereas when they decrease with scale ($H < 0$), they converge towards a mean state. To identify the various scaling regimes, we visually detected different sections with scaling behavior and fitted a linear segment to estimate the slope ($H$). These

exponents are shown as the slopes of the RMS values from the Haar fluctuation in a log-log plot. Figures 4 to 8 present the Haar fluctuation RMS for the aforementioned five latitude groups.

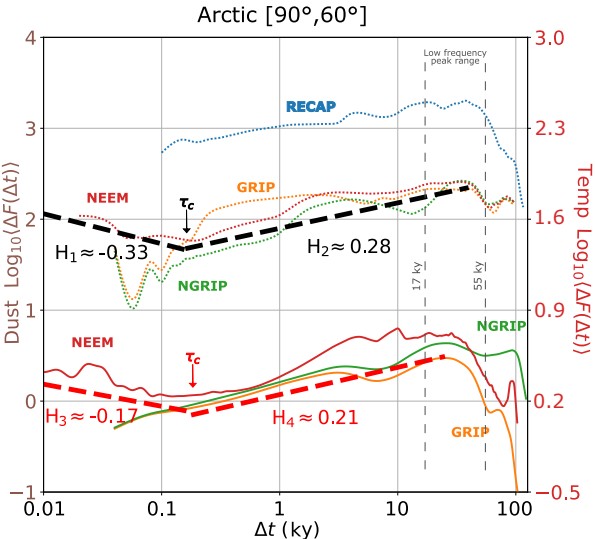

**Figure 4.** Haar fluctuation root mean square (RMS) values derived from the Arctic records at various scales, presented in log-log format. Bold letters along each curve denote the corresponding site names, identified by the data ID in Table 1. Units for each case vary and can be found in Table 1. Temperature proxies are displayed as continuous lines, and dust proxies as dotted lines. Thick dashed lines in black (dust) and red (temperature) were added as slope references, together with the estimation of the $H$ exponent. An arrow labeled with $\tau_c$ points to the estimated climate regime shift (fluctuation minimum). Vertical dashed gray lines indicate the multi-millennial scale range where fluctuations peak.

The Arctic Haar RMS fluctuations are shown in Figure 4. The finer resolution of the dust and temperature NEEM datasets allows us to identify the presence of an $H < 0$ regime (macroweather) and an $H > 0$ regime (climate), with a fluctuation minimum suggesting a climate regime change ($\tau_c$) between $\Delta t = 100$ and $300$ yr. Although the dust GRIP and NGRIP results

also show a minimum at higher frequencies, they have a slope greater than one, falling beyond the accurate range of the Haar analysis. Nevertheless, given the positive value of $H$ at multi-millennia scales and acknowledging that $H < 0$ at sufficiently high frequencies (the macroweather regime), we can deduce that $\tau_c$ is simply below the resolution covered. A similar scenario holds for the RECAP, GRIP and NGRIP temperature records. In addition, all sites exhibit higher fluctuations towards larger time scales (i.e., $H > 0$), peaking around 17 kyr and 55 kyr. As shown in the following figures, low-frequency peaks can be

observed across all latitudes.

The northern mid-latitude RMS fluctuations are shown in Figure 5. The terrestrial record from Xifeng (dotted blue) features a fluctuation minimum, suggesting a climate regime transition around 2 kyr. A low-frequency peak appears at time scales ranging between 12 kyr and 30 kyr. In the marine temperature records, a wide millennial-scale peak can also be identified in the Portugal site ($\Delta t \approx 3$ ky), as well as a low-frequency peak in the North Atlantic Ocean ($\Delta t \approx 30$ ky). In contrast,

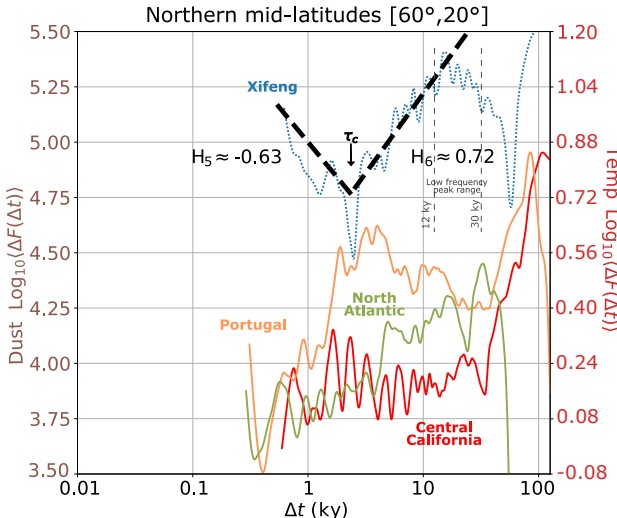

**Figure 5.** Haar RMS values derived from the northern mid-latitudes records. Temperature proxies are displayed as continuous lines, while dust proxies as dotted lines. Refer to Figure 4 for detailed annotations and references.

multi-millenial scale fluctuation amplitudes appear to plateau in the Central California site. None of the marine temperature sites exhibit a clear minimum in their fluctuations. Estimating $H$ for these records is not feasible because of the presence of oscillations at centennial to multi-millennial scales.

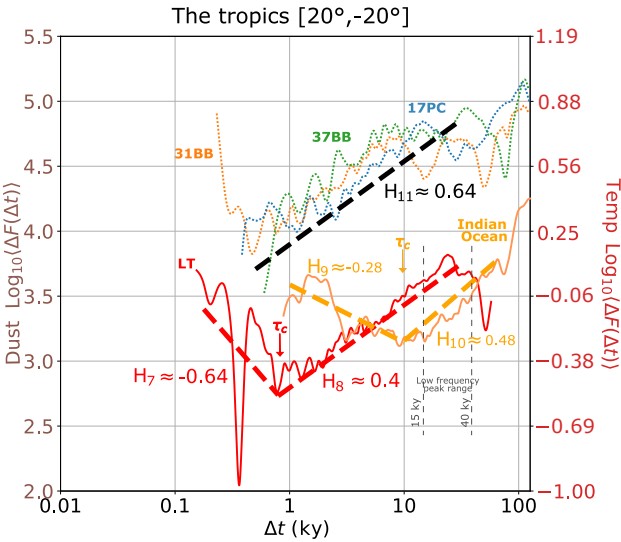

**Figure 6.** Haar RMS values derived from the tropical records. Temperature proxies are displayed as continuous lines, while dust proxies as dotted lines. Refer to Figure 4 for detailed annotations and references.

Figure 6 shows the RMS fluctuations from the tropics. Lake Tanganyika (LT) exhibits a fluctuation minimum ($\tau_c$) at time scales of approximately 600-1000 yr and a low frequency peak between 15 and 40 kyr. Conversely, the Indian Ocean features a $\tau_c$ close to 10 kyr, but a clear low-frequency peak cannot be identified. In contrast, marine dust records do not indicate an identifiable climate regime transition. Only 31BB (orange dotted line) shows a symmetry break at approximately $\Delta t = 500$ yr; nonetheless, as its slope on short timescales is less than the minimum detectable by Haar ($H < -1$), drawing any meaningful conclusions is not possible. However, because $H > 0$ at multi-millennial scales, we may suppose that $\tau_c$ is below the resolution of the data.

The PS75 marine record is the sole dataset located in the southern mid-latitudes. Despite oscillations at higher frequencies, Figure 7 shows a minimum ($\tau_c$) in its Haar fluctuation RMS within the time scales of 1.5 to 3 kyr. Beyond this range, the fluctuations increase with scale. A low-frequency peak is not detectable, which is consistent with the results from marine records in the northern mid-latitudes and the tropics.

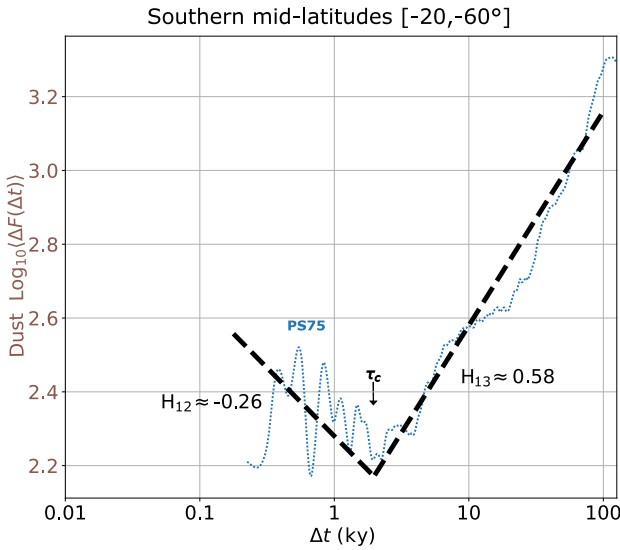

**Figure 7.** Haar RMS values derived from the southern mid-latitudes records. Temperature proxies are displayed as continuous lines, while dust proxies as dotted lines. Refer to Figure 4 for detailed annotations and references.

Finally, Figure 8 shows the Haar fluctuation RMS from the Antarctic records. We observe consistent results with similar scaling patterns across different sites and between dust and temperature proxies. The slopes of the reference lines indicate a minimum fluctuation between time scales of 200 and 750 yr in all cases, showing that the macro-weather-to-climate transition (our estimate of $\tau_c$) occurs at multi-centennial scales in the Antarctic. Furthermore, all datasets exhibit a low-frequency peak within the 15 to 45 kyr range.

Figure 9 illustrates the comparison of absolute amplitude differences among the available temperature data, measured in degrees Celsius. In this amplitude comparison, we exclusively utilise temperature data measured in degrees Celsius, thereby avoiding the need for calibration conversions, and mitigating potential biases in the analysis. Note that the amplitude calculated

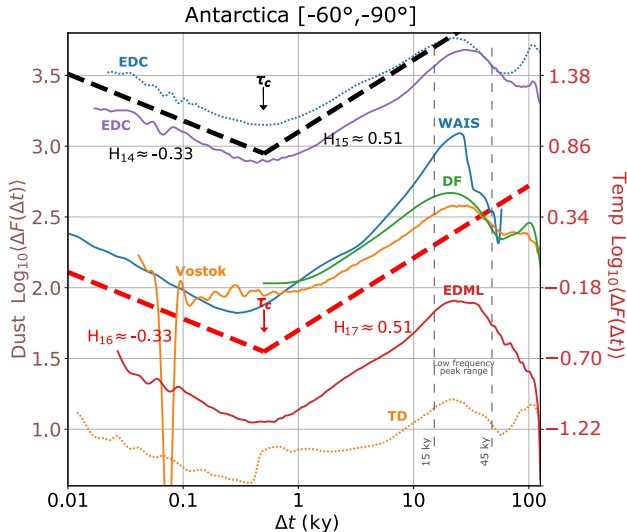

**Figure 8.** Haar RMS values derived from the Antactic records. Temperature proxies are displayed as continuous lines, while dust proxies as dotted lines. Refer to Figure 4 for detailed annotations and references.

here is approximately the total range of fluctuations. When comparing panels (a) to (c), it is evident that the amplitudes gradually decrease from the poles towards lower latitudes, going from 4 - 6°C at the poles to 1 - 2°C in the tropics, which is in line with the concept of polar amplification (Masson-Delmotte et al., 2006). However, it is worth noting that there are no data from the southern mid-latitudes in this analysis. Table 2 summarises the Haar statistics for the different latitudes.

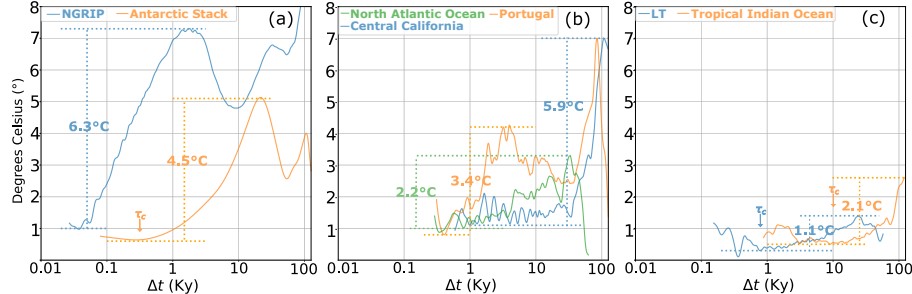

**Figure 9.** Absolute amplitude differences of the fluctuations for calibrated temperature datasets in degrees Celsius. Shown are (a) polar regions, (b) northern mid-latitudes, (c) tropics. The numbers accompanying the vertical lines estimate the fluctuation amplitude, equivalent to the difference between the peak found at multi-millennial scales minus the minimum found at multi-centennial or millennial scales. Some $\tau_c$ values are shown as reference according to the findings from Figures 4 to 8.

**Table 2.** Climate regime transition summary. *High frequency slope* denotes the $H$ exponent found at decadal/centennial time scales (e.g., $H_{14}$ or $H_{16}$ in Figure 8), *low frequency slope* denotes the $H$ exponent found at centennial/millennial time scales (e.g., $H_{15}$ or $H_{17}$ in Figure 8), and $\tau_c$ correspond to the transition timescale from *high frequency slope* to *low frequency slope*. Column *LF peak* shows the estimated range in kyr where a fluctuation maximum is attained, whereas column *Amplitude* refers to the difference between maximum and minimum fluctuations.

| Record | High frequency slope | Low frequency slope | $\tau_c$ (y) | LF peak (ky) | Amplitude (°C) |
|---|---|---|---|---|---|
| Arctic dust | -0.33 | 0.28 | 100-300 | 17-55 | - |
| Arctic temp. | -0.17 | 0.21 | 100-300 | 17-55 | 6.3 |
| Northern mid-latitudes Dust | -0.63 | 0.72 | 1,800-2,000 | 12-30 | - |
| Northern mid-latitudes temp. | - | >1 | - | - | 2-6 |
| Tropic dust | <-1 | 0.64 | - | - | - |
| Tropic temp. | -0.64&-0.28 | 0.4&0.48 | 600-10,000 | 15-40 | 1-2 |
| Southern mid-latitudes dust | -0.26 | 0.58 | 1,500-3,000 | - | - |
| Southern mid-latitudes temp. | - | - | - | - | - |
| Antarctica dust | -0.33 | 0.51 | 200-750 | 15-45 | - |
| Antarctica temp. | -0.33 | 0.51 | 200-750 | 15-45 | 4.5 |

## 4.2 Fluctuation correlations analysis

This section aims to quantify the interrelation between the poles and lower latitudes. We can estimate the pairwise time-dependent correlation coefficients using Equation (5). We perform this analysis separately on the temperature and dust records.

Figure 10 shows the fluctuation correlations of the temperature records. Here, we estimate 15 pole-pole correlations (i.e., the fluctuation correlations between Antarctic and Arctic records) and 40 pole-lower latitude correlations. Lake Tanganyika, the only terrestrial temperature record, is depicted independently of the other marine records. Thus, the light blue, red, and yellow areas highlight the pole-pole, pole-terrestrial, and pole-marine record correlations, respectively. Shaded areas cover one standard deviation from the mean, truncated by the upper limit of $\rho = 1$.

Figure 10 shows that all records exhibit lower fluctuation correlation values at high frequencies, which increase with scale. In particular, the poles attain $\rho = 0.75$ at $\Delta t = 10$ kyr (Figure 10(b)) and remain highly correlated across all scales. On the other hand, the correlation between the poles and lower latitude sites generally shows lower correlation values with higher standard deviation (wider yellow and red areas), indicating less synchronicity between themselves. It was unexpected that Lake Tanganyika (LT), despite being located in the tropics, correlates as rapidly as the poles, reaching $\rho = 0.75$ at similar scales. However, it shows a sharp drop towards $\Delta t = 50$ kyr, i.e., at scales of the length of the LT time series.

Similarly, Figure 11 shows the fluctuation correlations of the dust records. Here, we estimate eight pole-pole correlations and 30 pole-lower latitude correlations. Xifeng, the only terrestrial dust record, is depicted independently of the other marine records. Thus, the light blue, red, and yellow areas highlight the pole-pole, pole-terrestrial, and pole-marine record correlations, respectively.

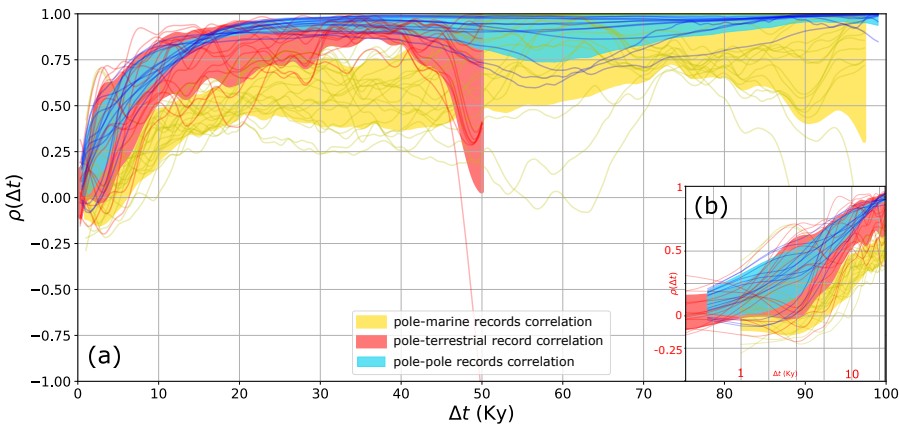

**Figure 10.** Fluctuation correlation of temperature archives. The light blue, red, and yellow areas highlight the pole-pole, pole-terrestrial, and pole-marine records correlations, respectively. Both bands cover one standard deviation from the mean, in both directions. Panel (a) displays the correlations between series up to $\Delta t = 100$ kyr, while panel (b) zooms in on only the first 20 kyr of $\Delta t$.

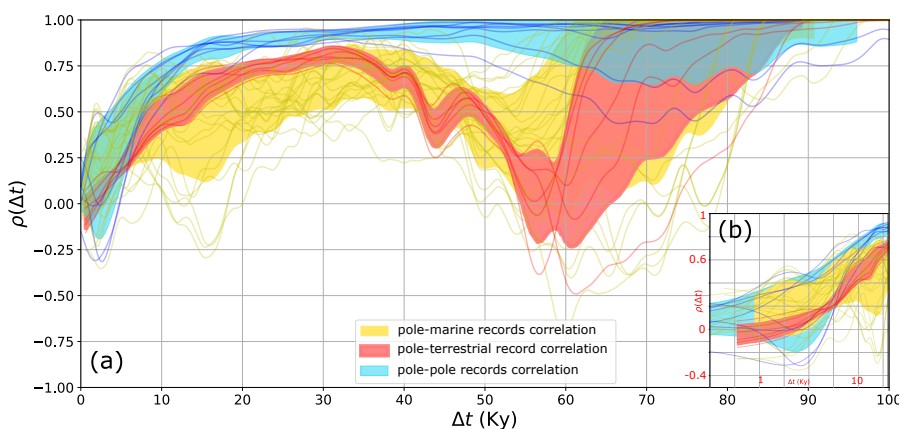

**Figure 11.** Fluctuation correlation of dust archives. The light blue, red, and yellow areas highlight the pole-pole, pole-terrestrial, and pole-marine records correlations, respectively. Both bands cover one standard deviation from the mean, in both directions. Panel (a) displays the correlations between series up to $\Delta t = 100$ kyr, while panel (b) zooms in on only the first 20 kyr of $\Delta t$.

In Figure 11, we observe a consistent pattern with our previous findings: all records exhibit lower fluctuation correlations
270    at high frequencies, gradually increasing to a nearly perfect positive correlation ($\rho = 1$) around scales of 100 kyr. Moreover, the correlation of polar records rises quickly with scale, reaching $\rho = 0.75$ at $\Delta t \approx 10$ kyr, and they remain strongly correlated with $\rho > 0.75$ over longer time scales. On the other hand, the pole-lower latitude relations also display stronger fluctuation correlations as $\Delta t$ increases, but at a slower pace (see Figure 11(b)) and never surpass the pole-pole correlation values.

Marine and terrestrial records seem to correlate similarly with the poles, both reaching a maximum of $\rho \approx 0.8$ at approxi-
mately $\Delta t$ = 35 kyr, then dropping rapidly from $\Delta t$ = 40 to 75 kyr. This drop is visible for both the pole-pole and pole-lower latitude correlations but is much more pronounced in the latter case, even reaching negative correlations for certain pairs of sites.

## 5  Discussion

In this study, we analysed global spatiotemporal atmospheric variability across annual to multi-millennial timescales using Haar fluctuations. Our results suggest that the transition time scale ($\tau_c$) from macroweather to climate is not consistent globally, but varies systematically according to latitude. While $\tau_c$ occurs at centennial to multi-centennial time-scales in polar areas ($\Delta t$ = 100-300 yr in the Arctic, $\Delta t$ = 200-750 yr in Antarctica), it occurs at longer multi-centennial to millennial time-scales at lower latitudes ($\Delta t$ = 600-10,000 yr in the tropics, $\Delta t$ = 1,500-3,000 yr in the mid-latitudes). Furthermore, almost all our $\tau_c$ estimations are around 1/100th of glacial cycle length scales, which is too short to be attributed to Milankovitch forcing, raising the question of what processes could be behind this transition.

Antarctica was the only group in which consistent results across all sites and variables were observed (Figure 8), probably because of the significant influence of similar atmospheric processes on Antarctic temperature and dust (Markle et al., 2018). Lovejoy and Lambert (2019) analysed high-resolution ice core data from the EDC dust series, estimating $\tau_c$ for eight glacial-interglacial phases, averaging approximately 500 years, which is consistent with our Antarctic $\tau_c$ estimation of 200-750 years. For the other latitude groups (Figures 4 - 7), the estimation of $\tau_c$ relied on data from only one or two sites, which introduced some uncertainty into the results. However, the overall consistency observed across different latitudes helps mitigate this uncertainty.

Fluctuations from terrestrial records (Figure 5 and 6) display a distinct low-frequency peak that appears to be centred around the precession and obliquity cycles. This peak is most probably related to variability in the East Asian and African Monsoon (Cheng et al., 2016; Tierney et al., 2008), but is absent in marine data, thus highlighting the impact of proxy-specific climate recording processes and underscoring why single-proxy analysis over mixed-proxy approaches should be preferred (Reschke et al., 2021). The diverse shapes of these peaks across records may be attributed to the varied influence of precession, obliquity, and eccentricity at different locations, as well as the limitations of the Haar fluctuation analysis, which tends to produce broad peaks rather than clearly identifying specific periodicities.

Another distinct characteristic of terrestrial records is that they do not match the $\tau_c$ calculated in marine records for their respective latitudinal zone. For example, the dust record from Xifeng in China (Figure 5) indicates a transition scale $\tau_c$ around 2 ka. Such a transition in not corroborated by the marine temperature records of the northern mid-latitudes that suggest a transition scale somewhere below the highest resolution of 0.5 ka. In the tropics, the terrestrial temperature record of Lake Tanganyika in East Africa suggests a transition scale $\tau_c$ around 600 years. In contrast, a marine temperature record from the tropical Indian Ocean suggests a transition around 10 ka, and no conclusive transition could be found in marine dust records

from the tropical Pacific within their resolution of about 0.5 ka. These discrepancies suggest that in addition to the large-scale latitudinal dependency of $\tau_c$ discussed below, some regional differences exist between marine and terrestrial environments.

By analysing the absolute amplitude differences of the fluctuations (Figure 9), we observed that regions with shorter $\tau_c$ values displayed larger fluctuation amplitude differences. Higher fluctuation amplitudes in polar areas and lower values in the mid-latitudes and tropics are a consequence of the polar amplification effect in both temperature and dust (Lambert et al., 2013). In Figure 12 we hypothesise that this polar amplification effect may be accompanied by a polar acceleration effect, whereby the transition from macroweather to climate occurs at shorter timescales at higher latitudes. One notable exception is the tropical records from Lake Tanganyika which displays a very small $\tau_c$ of about 0.6 ka, comparable with polar values. With the currently available records this hypothesis is weak because: 1. It is partly contradicted by the available data, 2. There are only dust records available for the mid-latitudes bands, and 3. There are only temperature records available for the tropical band. Still, the apparent coincidence between the two effects raises the interesting question of potential causality (in either direction) between the two. Additional reconstructions of both temperature (most urgently needed in mid-latitudes) and dust (most urgently needed in the tropics) are now needed to test the hypothesis.

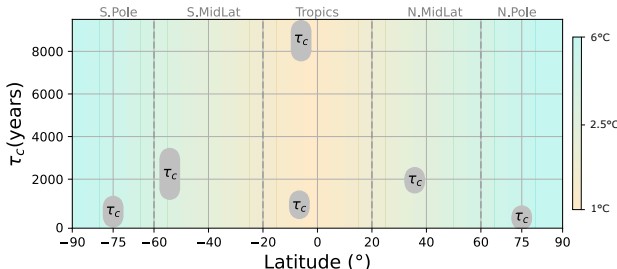

**Figure 12.** Schematic of the *Latitude-$\tau_c$-Amplitude* relation found considering both dust and temperature archives. Grey markers indicate the estimated range of $\tau_c$ values according to latitude, while the background color map, ranging from light blue to light red, represents the fluctuation amplitudes across latitudinal bands.

Fluctuation correlation analysis revealed that local variability dominates at high frequencies whereas global synchronisation appears at longer time scales (Figures 10 and 11). In particular, both the temperature and the dust results show that the poles synchronise within 10 kyr and remain highly correlated at longer timescales. The North Atlantic (where Greenland is located and where all our Arctic records are from) and the South Atlantic (which strongly influences Antarctic temperatures) are mechanistically linked through the bipolar sea-saw (Stocker and Johnsen, 2003). In addition to orbital forcing, these two regions are therefore also linked by this mechanism that produces synchronized multi-millennial variability described as Dansgaard/Oeschger Events in the north and Antarctic Isotopic Maxima in the South (EPICA Community Members et al., 2006). In contrast, fluctuation correlations between polar regions and lower latitudes show a slower increase towards multi-millennial scales, indicating no significant synchronising mechanism between the poles and the lower latitudes beyond the orbital forcing.

In temperature correlations, the terrestrial record of Lake Tanganyika (LT) unexpectedly exhibits a high and rapid correlation with the poles, despite its equatorial location. We note that this record also exhibits a very short $\tau_c$ similar in value to the polar

records. In contrast, the terrestrial dust record from Xifeng, China, synchronises with the poles at a pace similar to that of marine dust records at lower latitudes. It is therefore not clear whether the continental nature of the LT site plays a role in its synchronisation with the poles. We also note that LT's record is the shortest of all those selected in this study (approximately 60 kyr with 209 samples), reducing the confidence in the low-frequency range of this site's analysis, but not in the $\tau_c$ calculation.

Therefore, our results suggest that global synchronisation is primarily driven by the poles, as they exhibit shorter $\tau_c$ values and faster correlation among themselves compared to lower latitudes. This is consistent with the current paradigm that ice sheets in high latitudes were instrumental in driving global climate change (Abe-Ouchi et al., 2013). Conversely, as we approach the equator, the fluctuation amplitude diminishes and correlations decrease, resulting in longer transition time scales for fluctuations to participate in the glacial cycle.

## 6    Conclusions

In this study, we utilized Haar fluctuation analysis to explore the scaling properties of various paleotemperature and paleodust records. Our work underscores the importance of generating paleoclimatic datasets with both extensive coverage and high temporal resolution. Through the analysis of uniform and non-uniform time series, we confirmed the efficacy of Haar structure functions in assessing the scaling properties of paleoclimatic data, offering reliable scaling exponent estimates across a broad range of scales. Moreover, the straightforward interpretation, tied to fluctuations' behavior with scale changes, facilitates the characterization of climate transitions.

Our investigation of the $\tau_c$ parameter reveal latitudinal dependencies in the transition from macroweather to climate regimes. We observed centennial to multi-centennial transitions in polar regions versus longer multi-centennial to millennial transitions at lower latitudes, highlighting the intricate interplay between regional processes and global climatic influences. Furthermore, the $\tau_c$ parameter suggests the existence of a dominant forcing mechanism not aligned with Milankovitch's cycles (1/100th of glacial cycle length scales), prompting inquiries into its underlying causes.

Additionally, we found that the amplitude of fluctuations is greater at higher latitudes and decreases towards the tropics. This observation elucidates how $\tau_c$ shifts from shorter to longer scales and may explain why lower latitudes require more time to synchronize with the poles, as the modulated signal propagating from the poles to the equator weakens. Moreover, this finding may help reconcile some Holocene results, suggesting that the macroweather-climate transition occurs at millennial rather than centennial scales, as Holocene data are often collected in mid-latitudes or tropics, where longer transition scales are expected (e.g., Lovejoy and Schertzer, 2013).

Fluctuation correlation analysis revealed that poles synchronise faster than lower latitudes, achieving full correlation before half a cycle (50 kyr), indicating synchronous behaviour during glacial periods. This underscores the significant impact of Dansgaard-Oeschger/AIM events and orbital forcing in polar regions. In contrast, the co-variability between polar regions and lower latitudes is less influenced by millennial-scale events and is more concentrated around orbital periodicities, with a distinct drop in dust correlation in the band between obliquity and eccentricity.

In summary, the latitudinal dependency in $\tau_c$ and the observed signal propagation from the poles to the equator underscore the significance of polar regions in influencing global climatic variability. Thus, our findings represent the first empirical evidence supporting the hypothesis of the pivotal role of the poles as climate change drivers.

Future studies can enhance fluctuation estimation robustness by comparing or averaging results across different glacial cycles. Additionally, assessing lagged cross-correlations between poles, mid-latitudes, and tropics may clarify directional causality, albeit without conclusive evidence. Furthermore, addressing the current scarcity of high-resolution time series from mid-latitudes and the tropics is crucial for improving the characterisation of geographic variability in glacial-interglacial cycles at these latitudes.

*Code and data availability.* The data are available as supplement of the references listed in Table 1. A repository containing the Python function to estimate Haar fluctuations on non-equidistant datasets is available at https://github.com/NicoAcunaR/Non-equidistant_Haar_fluctuations.

## Appendix A: Haar fluctuations on non-equidistant data

The definition of Haar fluctuation in Equation (1) assumes a uniform timestep. Here, we follow an approach similar to Lovejoy (2015, App. B) and extend the Haar analysis to non-uniform time series, as necessary, to accurately analyze the datasets in our study. First, let us assume that there are $N$ measurements of process $T(t_i)$, where $(t_i)_{i \in [1,N]}$ are irregular time samplings. Then, we define the cumulative sum $S_i$ as

$$S_i = \sum_{j \leq i} T(t_j). \tag{A1}$$

By taking an index $j$ and an even number $k$, we can define $[t_j, t_{j+k}]$ as the interval where the fluctuation $\Delta T_{j,k}$ is estimated. To do so, we determine the sum of $T(t_i)$ over the first and second halves of the interval by

$$\Delta S^{(1)} = S_{j+k/2} - S_j \quad , \quad \Delta S^{(2)} = S_{j+k} - S_{j+k/2}, \tag{A2}$$

and compute the average of the first half minus the average of the second half of the $[t_j, t_{j+k}]$ interval as:

$$\Delta T_{j,k} = 2 \left( \frac{\Delta S^{(1)}}{t_{j+k/2} - t_j} - \frac{\Delta S^{(2)}}{t_{j+k} - t_{j+k/2}} \right). \tag{A3}$$

Here, the canonical factor 2 multiplied by the difference is a calibration parameter necessary to expand the working range from the anomaly ($H < 0$) and differences ($H > 0$) to the Haar range $-1 \leq H \leq 1$, see Lovejoy (2023).

This definition works for any sampling rate, but its accuracy deteriorates when the first and second halves of the interval cover disproportionate time durations. Hence, we included quality control in the uniformity of the intervals. For this purpose, we define the nonuniformity of the interval $\Delta t$ as the ratio between the time samples:

$$\epsilon = \frac{t_{j+k/2} - t_j}{t_{j+k} - t_j}. \tag{A4}$$

Then, we define a threshold $\epsilon_{\min}$ and include only Haar fluctuations on intervals that satisfy $\epsilon_{\min} \leq \epsilon \leq (1 - \epsilon_{\min})$. Decreasing $\epsilon_{\min}$ implies a precision loss, and hence smoothens the resulting curve. Differently, taking $\epsilon_{\min}$ too close to 1/2 (uniform sampling case) can result in the exclusion of too many fluctuations, with the consequence that the statistics will be poor. For this study, we fixed $\epsilon_{\min}$ at 1/4.

The first-order Haar structural function in Equation (2) can be readily applied to the definition of non-uniform intervals. Furthermore, the second-order structure function can be used to estimate the Haar fluctuation root-mean-square (RMS) value as

$$\langle |\Delta F(\Delta t)|^2 \rangle^{1/2} \propto \Delta t^{\xi(2)/2} = \Delta t^{H - K(2)/2}, \tag{A5}$$

as per Lovejoy and Lambert (2019). This statistic fully characterises the scaling of Gaussian processes and provides important insights for non-Gaussian datasets. Thus, we used the RMS fluctuation as an approximation to the absolute mean, that is, $H \approx \xi(2)/2$, in this study.

*Author contributions.* All authors participated in the conceptualization of the research and the methodology. NA developed the software and visualization and conducted the formal analysis and investigation. EV, FL and SL provided supervision. NA prepared the original draft, and all authors contributed to reviewing and editing the final paper.

*Competing interests.* The authors declare that they have no conflict of interest.

*Acknowledgements.* This study was undertaken by members of the CVAS working group of the Past Global Changes (PAGES) Global Research association. The study was funded by ANID/Fondecyt/1231682 and ANID/Fondecyt/12316821221220.

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
