# Peer review of "Geographic variability of dust and temperature in climate scaling regimes over the Last Glacial Cycle."

_EGUsphere, 2023_

## Author Comment (AC1)

**Response to Reviewer Comments**

Dear Dr. Anders Svensson and Anonymous Referees,

Thank you for investing your time in reviewing our manuscript, titled "Geographic variability of dust and temperature in climate scaling regimes over the Last Glacial Cycle." The insightful feedback significantly contributed to the improvement of our paper, notably by highlighting outdated or inaccurate data in our database.

In response to the reviewers' comments and queries, we have addressed each point comprehensively. Below, you will find our responses, presented in regular font, alongside the reviewers' comments, denoted in *italics*.

**Anders Svensson review**

*Major Comments:*

1) *To put a bit more focus on the applied datasets, it would be great to see all of the applied records shown on a common time scale in a figure*:
   We have adjusted Figure 3, now depicting the last 130 ky for all the records under consideration.

2) *Another important point to mention is the time scales you are applying for each record*:
   We have incorporated a new column into Table 1 to detail the time scales employed.

3) *It would also be helpful to show the time resolution of the applied records throughout the 120 ka period you investigate*:
   Due to the nonuniform sampling in many datasets, representing display resolution on a single figure proves challenging. Our primary focus lies in highlighting the nonuniformity of the datasets rather than emphasizing specific resolutions. Therefore, we have concluded that Figure 2, illustrating cumulative age versus cumulative number of samples, best captures this aspect. Additionally, we maintain a mean resolution record in Table 1.

4) *An alternative interpretation of Figure 5 would be that the variability in correlation among records does not reflect climate variability, but rather that time scales of the applied records tend to disagree more the longer the record is, in particular between ice cores and lower latitude records*:
   We delved deeper into this point with the goal of employing synchronous timescales. While it wasn't feasible to transform all datasets to a common timescale, our priority was to use AICC12 and GICC05 timescales whenever possible. Most polar records are therefore on a common timescale now and the shape of the curve is still the same. In addition, a progressive disagreement over time would produce a tendency for all timescales (but especially for the high frequencies) towards an "average" correlation, which is not observed in our results. This suggests that the

observed strong variability in correlation among records is not attributable to disagreements in timescales.

5) *Figure 6 Very strange that the analysis gives so different results for GRIP and NGRIP as the records are practically identical:*
Thank you very much for pointing this out. We inadvertently used an erroneous GRIP dataset that significantly differs from NGRIP. Acknowledging this oversight, we have consulted Rasmussen et al. (2014) and rectified our error by using both the GRIP and NGRIP datasets from that study with a uniform resolution of 20 years.

6) *Around line 265, a discussion of DO and AIM events suddenly starts without previous introduction. Likewise, the knowledge about the origin of ice core dust is relevant for your interpretation and should have been mentioned earlier. I think it would be relevant to introduce to those topics in the introduction section and then to comment on them in the discussion and compare to the results of your analysis?*
We have expanded the introduction to include background on these two topics.

7) *Statement in lines 266-268 about polar amplification needs to be supported:*
We added a reference to Lambert et al. 2013: "The role of mineral dust aerosols in polar temperature amplification.".

8) *Statement line 278 about the importance of ice sheet height for temperature is rather incomplete. What about DO-events, glacial-interglacial periods, and sea ice extent in the North Atlantic? Are those factors not important for the site temperature on the ice sheet?:*
The discussion has been reworked based on updated results and reviewer suggestions. This particular paragraph has been removed.

**Anonymous referee RC2**

*Major Comments:*

1) *Clarify whether the specific methodological approach is novel or the technique was already presented in previous papers (beyond the application to a broader/different dataset):*
The specific methodological approach is not completely novel. In this study, we followed the theoretical work of Lovejoy and Lambert (2019) and improved on an existing simpler version of the Haar algorithm. What sets our approach apart is the consolidation of various technical aspects within a single paper and the incorporation of datasets with uniform and nonuniform sampling. We adjusted the narrative to clarify this point throughout the paper.

2) *Figure 2 and related text: it is not very clear to me what this variable represents, please give a clearer definition:*

We modified the layers of Figure 2 and its caption from 'proportional age vs sample' to 'cumulative samples vs cumulative age.' Additionally, we added a brief explanatory line to facilitate interpretation.

3) *Why does delta-t in Figure 4d,e have negative values? What does that represent?*:

Initially, all x-axes of Figures 4 (now Figures 4 to 8) were presented in l$og10(delta\_t)$ rather than $delta\_t$. Given that $delta\_t$ is measured in thousands of years, the log10 axis spanned from -2 (10 years) to 2 (100,000 years). To enhance readability, we opted to directly display the x-axis as $delta\_t$. This adjustment expands the coverage from 0.01 to 100 thousand years. The negative values observed in old Figure 4d and e (now changed to Figures 4 and 5) were an oversight on our part.

4) *Line 177. "low-frequency peak somewhere between 17.8 to 31.6 kyr" , where can we see that?:*

To highlight the observation of low frequency peaks, we have added vertical dash lines to Figure 4 to 8 depicting it. These peaks were also added to Table 2.

5) *Line 247. Concerning NGRIP, I thought dust and d18O were essentially co-varying on millennial time scales, so I would expect to read something different at least concerning dust on these time scales. Could you comment on that?:*

That is correct. As can be seen in (revised) Figure 11, dust and temperature in Greenland are highly anticorrelated at timescales longer than 2000 years. With the updated NEEM data we now identify identical transition scales in both dust and temperature records. These results are now discussed in the updated discussion section.

6) *Sections 4 and 5: spend more time describing and discussing the meaning of your findings:*

We fully modified sections 4 and 5 to delve deeper into the meaning of our findings.

**Anonymous referee RC3**

*Major Comments:*

1) *A more robust comparison to other glacial-age interpretations would strengthen the authors findings and interpretations*:

Most other works on climate scaling has focused on the Holocene when data is comparatively abundant. This is the first study that systematically analyzes glacial-age records. We compare our Antarctica findings with those of Lovejoy and Lambert (2019), who averaged and analyzed results from 8 glacial cycles in EDC dust time series. Our results align consistently with theirs.

---

## Author Response (AR2)

**Technical corrections**

Dear Laurie Menviel,

In response to the reviewers' comments and queries, we have addressed each point comprehensively. Below, you will find our responses, presented in regular font, alongside the reviewers' comments, denoted in *italics*.

*1-Could you provide some possible reasons as to why the transition from macro weather to climate is shorter at high than low latitudes?:*

- We discuss this problem in lines 311-318, but we limit our discussion in this manuscript to naming and describing the phenomenon. We don't want to guess as to the reasons behind it.

*2-Please consider adding in the Introduction and Discussion a few sentences on the physical processes and drivers that underpin your study of the climate regime change:*

- In terms of climate variability at different timescales, these processes are already mentioned in the introduction in lines 88-105. We have some general information on the proxies we use in lines 106-119, also in the introduction. Since we consider timeseries from different archives and sites, a description of physical processes in each case would take too much space. In the discussion, various processes and drivers are mentioned in lines 319-338.

*L. 62-63: "averaging more and more seems to converge", can you please rephrase by being more explicit?*

- We rephrased this to "by increasing the timescale the average converges more and more".

*L. 76-77: I do not follow the logic here as to why all climate variability has to be linked to Milankovitch.*

- In our text we indicate exactly this point, that the timescales involved for tau_c are too short to be related to Milankovitch and that some other currently unknown forcing mechanism has to be at play.

*L. 81: "thought" iof "though"*

- Corrected.

*L. 93-95: Please rephrase that sentence as its odd structure does not make it correct*

- We broke the sentence in two to make it clearer.

*L. 100-103: I do not understand why the transition between macro weather and climate can only be studied in empirical records and not modelling studies.*

- In the text we mentioned that because the models cannot replicate the large low-frequency variability of the climate, the transition can only RELIABLY be investigated in empirical records. Of course, the same can be done in climate simulations, but taking in mind the relative weakness of climate models at these time scales.

*L. 123: please rephrase "understanding of the whole"*

- Rephrased to "spatiotemporal analysis".

*L. 197-198: Should this sentence be moved to the Methods?*

- We moved this sentence to the description of Figure 9, since we think it fits better in the results section than in the more general methods section.

*L. 210 and other instances, please use the present tense in the Results section*

- Corrected.